# Biofilm Formation, Motility, and Virulence of *Listeria monocytogenes* Are Reduced by Deletion of the Gene *lmo0159*, a Novel Listerial LPXTG Surface Protein

**DOI:** 10.3390/microorganisms12071354

**Published:** 2024-07-02

**Authors:** Weidi Shi, Qiwen Zhang, Honghuan Li, Dongdong Du, Xun Ma, Jing Wang, Jianjun Jiang, Caixia Liu, Lijun Kou, Jingjing Ren

**Affiliations:** 1College of Animal Science and Technology, Shihezi University, Shihezi 832000, China; shiweidi1127@163.com (W.S.); zhangqiwen0056@126.com (Q.Z.); lhh121004@126.com (H.L.); jiangjianjun@shzu.edu.cn (J.J.); liucaixia0402@163.com (C.L.); klj175@sina.com (L.K.); renjingjing0524@163.com (J.R.); 2Analysis and Testing Center, Xinjiang Academy of Agriculture and Reclamation Science, Shihezi 832000, China; dudongdong0858@126.com; 3Key Laboratory of Control and Prevention of Animal Disease, Xinjiang Production & Construction, Shihezi 832000, China

**Keywords:** foodborne pathogen, *L. monocytogenes*, deletion mutant, bacteria–host interactions, RT-PCR

## Abstract

Listeria monocytogenes (*L. monocytogenes*) is a foodborne pathogen that causes listeriosis in humans and other animals. Surface proteins with the LPXTG motif have important roles in the virulence of *L. monocytogenes*. Lmo0159 is one such protein, but little is known about its role in *L. monocytogenes* virulence, motility, and biofilm formation. Here, we constructed and characterized a deletion mutant of *lmo0159* (∆*lmo0159*). We analyzed not only the capacity of biofilm formation, motility, attachment, and intracellular growth in different cell types but also LD_50_; bacterial load in mice’s liver, spleen, and brain; expression of virulence genes; and survival time of mice after challenge. The results showed that the cross-linking density of the biofilm of ∆*lmo0159* strain was lower than that of WT by microscopic examination. The expression of biofilm-formation and virulence genes also decreased in the biofilm state. Subsequently, the growth and motility of ∆*lmo0159* in the culture medium were enhanced. Conversely, the growth and motility of *L. monocytogenes* were attenuated by ∆*lmo0159* at both the cellular and mouse levels. At the cellular level, ∆*lmo0159* reduced plaque size; accelerated scratch healing; and attenuated the efficiency of adhesion, invasion, and intracellular proliferation in swine intestinal epithelial cells (SIEC), RAW264.7, mouse-brain microvascular endothelial cells (mBMEC), and human-brain microvascular endothelial cells (hCMEC/D3). The expression of virulence genes was also inhibited. At the mouse level, the LD_50_ of the ∆*lmo0159* strain was 10^0.97^ times higher than that of the WT strain. The bacterial load of the ∆*lmo0159* strain in the liver and spleen was lower than that of the WT strain. In a mouse model of intraperitoneal infection, the deletion of the *lmo0159* gene significantly prolonged the survival time of the mice, suggesting that the *lmo0159* deletion mutant also exhibited reduced virulence. Thus, our study identified *lmo0159* as a novel virulence factor among *L. monocytogenes* LPXTG proteins.

## 1. Introduction

*Listeria monocytogenes* (*L. monocytogenes*) is an opportunistic, intracellular pathogen that causes foodborne infections in humans and animals [1]. Immunocompromised individuals, such as newborns and pregnant women, are at particular risk for listeriosis, the clinical manifestations of which include meningitis, meningoencephalitis, septicemia, abortion, and a mortality rate as high as 30% [2,3].

*L. monocytogenes* can cross several nearly impenetrable host barriers, including the intestinal, blood–brain, and fetoplacental barriers [4]. It can invade and multiply within macrophages, hepatocytes, endothelial, and epithelial cells and then spread directly from cell to cell, without exposure to the outside environment, thus persistently infecting host cells and evading the host’s adaptive immune response [5]. Each step of the infection process of *L. monocytogenes* depends on the production of virulence factors, and the LPXTG surface proteins have been shown to play critical roles in the adaptability of *L. monocytogenes* to environmental stress, adhesion, invasion, and immune evasion [6,7]. The first LPXTG protein identified in *L. monocytogenes* was InlA. The InlA–E-cadherin interaction promotes bacterial entry into epithelial cells through specific binding mediate, thereby facilitating the movement of *L. monocytogenes* across the intestinal barrier [8,9]. Gp96 is a receptor for a novel *L. monocytogenes* virulence factor, Vip, a surface protein, and the interaction of Vip with Gp96 promotes bacterial invasion in some mammalian cells. The deletion mutant of the Vip gene significantly decreases the virulence of *L. monocytogenes* [10]. InlH, an internalin belonging to the LPXTG family, contributes to *L. monocytogenes* evasion of host defense in host defenses by specifically downregulating the IL-6 response. The InlH mutant reduces the virulence of *L. monocytogenes* in mice [11]. Lmo0159 has been documented as a member of the LPXTG family, but its function and mechanisms are still largely unknown, including the biofilm-formation ability and the virulence of *L. monocytogenes* in vivo.

Biofilm formation facilitates bacterial colonization and has been implicated in reduced susceptibility to the host immune response. This study investigated the role of Lmo0159, a novel Listerial LPXTG surface protein, on the *L. monocytogenes* virulence and biofilm formation. We describe for the first time the ability of mutant strains to adhere to, invade, survive intracellularly in, and proliferate in various cell lines; determine virulence in vivo; and form biofilms. The data suggest that the LPXTG surface protein encoded by *lmo0159* is a novel virulence factor of *L. monocytogenes* that may facilitate entry into eukaryotic cells and biofilm formation. In addition, the described findings may significantly impact the understanding of the molecular process of *L. monocytogenes* pathogenesis.

## 2. Materials and Methods

### 2.1. Bacterial Strains, Plasmids, and Primers

The LM90SB2 (WT) strain belonging to the serotype 4b was isolated from sheep with listeriosis by Professor Xun Ma in Xinjiang Production and Construction Corps in 2000 and cryopreserved by the Microbiology Laboratory of College of Animal Science and Technology, Shihezi University. The complete genome sequence data of the isolate were deposited in the National Center for Biotechnology Information (NCBI), GenBank, accession number GCA_028867355.1. The strains were grown at 37 °C on BHI agar. The temperature-sensitive shuttle plasmid pKSV7 [12] was kindly provided by Professor Weihuan Fang of the Laboratory of Molecular Microbiology and Food Safety, School of Animal Sciences, Zhejiang University. The pMD19-T (Simple) vector was purchased from TaKaRa Company, Kusatsu, Japan. The primers used in this study are listed in Table 1.

### 2.2. Cells and Cell Culture

The human cerebral microvascular endothelial cell line (hCMEC/D3) was purchased from BeNa Culture Collection, Xinyang, China; the murine brain microvascular endothelial cell line (mBMEC) was purchased from Guangzhou Jennio Biotech Technology, Guangzhou, China; the swine intestinal epithelial cell (SIEC) line and the mouse macrophage cell line (RAW264.7) were preserved in our laboratory; NCTC clone 929 (L929) was purchased from Procell Life Science&Technology Co., Ltd, Wuhan, China. All cells were maintained at 37 °C in a humidified atmosphere of 5% CO_2_. For the SIEC line and RAW264.7 cells were grown in Dulbecco modified Eagle’s medium (DMEM) (Gibco, Grand Island, NY, USA) supplemented with 10% fetal bovine serum (FBS) (Excell Bio, Shanghai, China). mBMECs were grown in DMEM supplemented with 15% FBS. hCMEC/D3 cells were grown in endothelial cell medium (ECM) (Gibco, Grand Island, NY, USA) supplemented with 5% FBS and 1% endothelial cell growth supplement (ECGS) (Gibco, Grand Island, NY, USA).

### 2.3. Experimental Design

The experimental design of the current study is described below and illustrated in Figure 1. First, a *lmo0159* deletion mutant (∆*lmo0159*) was constructed via the homologous recombination technique. In an in vitro study, motility was detected in plaque size and plate diffusion size. Then, the biofilm-formation ability was determined by staining, micrographs, and biofilm gene detection. To evaluate the role of ∆*lmo0159* on intracellular bacterial survival, we used the adhesion, invasion, and intracellular proliferation assays in cell models, including SIEC, RAW264.7, mBMEC, and hCMEC/D3. Cell migration by *L. monocytogenes* was measured by scratch-healing assay. In vivo, each experimental group consisted of eight mice. The virulence level was evaluated by the mouse survival rate, LD_50_, tissue (liver, spleen, and brain) bacterial load, and expression of virulence genes.

### 2.4. Construction of the ∆lmo0159 Mutants Strain

The *lmo0159* gene of the WT strain was deleted as follows. The flanking regions (A1 (388bp) and A2 (409bp) homology arm fragments) of the target gene *lmo0159* were amplified by PCR with the primer pairs of 0159-F1-R1 and 0159-F2-R2. After purifying the PCR product from agarose gels, the A1 and A2 homology arms are templates. The fusion fragment of the *lmo0159*-deleted gene (∆*lmo0159*) was obtained by SOE-PCR using 0159-F1 and 0159-R2 as primers (Table 1). The ∆*lmo0159* fragments were cloned into the temperature-sensitive shuttle plasmid pKSV7. The resulting plasmid was transformed into the WT strain and plated on BHI, with chloramphenicol at 30 °C for 48 h to force the integration of the plasmid into the bacterial chromosome by a single crossover. Positive pKSV7-∆*lmo0159* colonies were selected by PCR and then subcultured for 25 generations at 41 °C in a medium containing 10 μg/mL chloramphenicol to ensure the integration of the pKSV7 into the WT chromosome via homologous recombination. These so-called integration mutants were then grown in a chloramphenicol-free BHI medium culture at a permissive temperature (30 °C) conducive to selecting for plasmid excision and loss of chloramphenicol resistance. Chloramphenicol-sensitive recombinant ∆*lmo0159* was selected by PCR, and sequencing was used to identify ∆*lmo0159* to ensure that the *lmo0159* gene was deleted.

### 2.5. Growth Assay

WT and ∆*lmo0159* strains were grown in BHI medium at 37 °C overnight, and the optical density at 600 nm (OD_600_) of the cultures was measured using an enzyme-labeled instrument (BioTek, Winooski, VT, USA). The OD_600_ of the cultures was adjusted to 0.5 using BHI medium. The adjusted cultures were inoculated at 1:100 into fresh BHI medium. Data for growth curves were collected by measuring changes in OD_600_ at two-hour intervals over 12 h, using a microplate reader. Optical density values are the means of data from three independent experiments.

### 2.6. Biofilm-Formation Assay

The crystal violet staining assay and light microscopy were used to detect the biofilm-formation ability of *L. monocytogenes* at different time intervals. Four clean, grease-free glass slides were prepared in a 90 mm petri dish. Petri dishes were autoclaved. The overnight cultures of WT and ∆*lmo0159* strains were split 1:10 and cultured until an OD_600_ of 0.2 was reached. Then, 8 mL of the cultured bacteria was aliquoted into each petri dish, and the respective petri dish sets were incubated at 37 °C for 8, 12, 24, and 48 h. Slides were removed and washed three times with PBS to remove the unattached cells. Slides were allowed to be air-dried at 55 °C for 30 min and then stained with 0.1% crystal violet solution for 30 min. Slides were washed thrice with PBS, air-dried for 30 min at room temperature, and photographed with an inverted microscope.

Quantitative biofilm-formation assays were performed as previously described, with minor modifications [13]. Briefly, aliquots of WT and ∆*lmo0159* strains were inoculated in 96-well plates, with 150 μL per well. Biofilm formation was performed as described above after 8, 12, 24, and 48 h of inoculation at 37 °C. Deposited crystal violet was solubilized by 150 μL of 95% ethanol for 30 min. Cell turbidity was monitored by a microtiter plate reader at an optical density of 570 nm (OD_570_) and recorded at various time intervals. The final OD_570_ for the turbidity and crystal violet was calculated from the average OD_570_ of eight test wells. All the experiments were repeated three times.

### 2.7. Motility Assay

#### 2.7.1. Swimming Motility Assay

Established methods [14,15] were followed. WT and ∆*lmo0159* strains were incubated overnight at 37 °C, and the OD_600_ value was adjusted to 0.6. TSA semi-solid medium (0.25% AGAR, 2% sodium chloride, and 1.5% tryptone) was stabbed after dipping the front quarter of a sterilized toothpick into the bacterial solution and then placed at 25 °C for static culture. The colony size and umbrella motility circles of *L. monocytogenes* were observed within 48 h.

#### 2.7.2. Plaque Forming Assay

The methods used [16] were described previously. The hCMEC/D3 and L929 cell lines were plated on uncoated 12-well culture dishes at a density of 1 × 10^5^ cells·well^−1^ and incubated until they reached 95% confluence. These cells were infected as described above, with the *L. monocytogenes*-challenged hCMEC/D3 at a MOI of 1:1 and L929 cells at a MOI of 0.1:1. After 60 min of incubation at 37 °C in a 5% CO_2_ atmosphere, the wells were washed twice with PBS and covered with fresh cell culture medium containing 100 μg/mL of gentamicin to kill extracellular bacteria. After 1 h, the medium was changed to a suspension (1:1) containing 10 μg/mL of gentamicin to which methylcellulose and cell culture medium were added. After 60 h, the cells were fixed with 4% formaldehyde and stained with 1% crystal violet.

### 2.8. Scratch Assay

The experiment was performed according to a previously described method [17]. Briefly, the L929 cell line was plated on uncoated 12-well culture dishes at a density of 1 × 10^5^ cells-well^−1^ and incubated until it reached 95% confluence. Cells were incubated in the presence of WT and ∆*lmo0159* for 8 h. Cell monolayers were scratched manually with a pipette tip. Cell monolayers were then washed twice with sterile PBS to remove cellular debris. 

Analysis of the cell-free area at 0 and 24 h was performed using ImageJ software (version 1.52a, National Institutes of Health, Bethesda, MD, USA). The cell-covered area was plotted to determine the rate of wound closure. The data were then normalized, and the result was expressed as the percentage of change in cell-front velocity in cell-free areas. The migration rate was calculated as the average relative difference between the cell-free areas using six independent experiments.

### 2.9. Adhesion and Invasion Assays

The methods described in previous studies [18] were followed. The *L. monocytogenes* strains were grown overnight in BHI medium at 37 °C until the optical density at 600 nm (OD_600_) approached approximately 0.8 to 1. After washing twice with phosphate-buffered saline (PBS, pH7.4), the bacteria were diluted in DMEM and then added to hCMEC/D3, mBMECs, and SIECs at a MOI of 100:1 and RAW264.7 cells at a MOI of 10:1. After bacterial suspensions were added to eukaryotic cells and incubated at 37 °C in a 5% CO_2_ atmosphere for 1 h, the cells were washed three times with PBS to remove the non-adhered bacteria. After washing, the cells were harvested by lysis in 0.2% Triton X-100 and 0.25% trypsin, and the number of viable bacteria released from the cells was determined by serial dilution plating on BHI agar plates. Invasion rates were determined by the gentamicin protection method. Eukaryotic cells were infected as above, washed three times with PBS buffer, and noninvasive bacteria were killed by adding 100 μg/mL gentamicin for 1 h. After the cells were washed three times with PBS described above, infected cells were lysed to determine the number of viable intracellular bacteria. The experiment was repeated three times.

### 2.10. Intracellular Proliferation Assay

These cells were infected as described above after incubation for 60 min at 37 °C in a 5% CO_2_ atmosphere. The wells were washed twice with PBS and covered with fresh cell culture medium containing 100 μg/mL gentamicin to kill extracellular bacteria [18]. This time was considered 0 h for the intracellular growth assay. After 1 h, the cell culture medium containing 10 μg/mL gentamicin was replaced. As described above, the number of viable bacteria was determined 2, 4, 6, 8, 10, and 12 h later. The experiment was repeated three times.

### 2.11. The Mice Survival Rate Assay and the Detection of 50% Lethal Dose (LD_50_) 

Eight 6-to 8-week-old specific pathogen-free female BALB/c mice were used for each experiment and injected intraperitoneally with 0.5 mL of 1.2 × 10^7^ CFU WT and ∆*lmo0159* grown in BHI for 18 h at 37 °C, washed twice with PBS, and resuspended in PBS. The control groups were injected intraperitoneally with PBS instead. Mice were inoculated intraperitoneally, the number of deaths was recorded, and the corresponding eight-day survival curves were calculated.

The WT and ∆*lmo0159* strains were inoculated into BHI (5 mL) and cultured at 37 °C to an OD_600_ of 0.40. Each culture was centrifuged (at 5000 rpm and 4 °C), the supernatant was discarded, and the bacteria were collected. Bacteria were washed twice with PBS, suspended in PBS, and serially diluted 10-fold. Six-to-eight-week-old female BALB/c were randomly assigned to the WT or ∆*lmo0159* groups. Each group had five subgroups, with eight mice in each subgroup. Mice in each subgroup were injected intraperitoneally with 0.5 mL of serially diluted bacterial suspension. The mice were continuously observed for 10 days, and the number of deaths was counted. The LD_50_ of WT and ∆*lmo0159* strains in mice was determined by the Kerber method [18].

### 2.12. Bacterial Loads in Mice Livers, Spleens, and Brains

Six-to-eight-week-old female BALB/c mice were used in groups of twelve mice for each bacterial strain. The experiment was repeated three times. Mice were infected intraperitoneally with 0.5 mL of WT- and ∆*lmo0159*-strain bacterial suspension. Mice (n = 3 per group) were sacrificed at 24, 48, and 72 h post-challenge. Livers, spleens, and brains were dissected under a sterile condition, and CFU counts were determined after serial dilutions of organ homogenates were plated on BHI agar plates and incubated at 37 °C [18].

### 2.13. RT-PCR Assays

WT and ∆*lmo0159* strains were grown on the respective petri dishes and incubated at 37 °C for 48 h. Total RNA from the cultured strains was isolated using Transzol UP reagent (Transgen, Beijing, China) according to the manufacturer’s instructions. One microgram of RNA was reverse-transcribed into cDNA using the PerfectStart Uni RT&qPCR Kit (Transgen Biotech, Beijing, China), and the primer sets for RT-PCR are shown in Table 2. Virulence gene primer sets for *prfA*, *plcA*, *hly*, *mpl*, *actA*, *plcB*, *inlA*, *inlB*, and *inlC* with *gyrB* as the housekeeping gene and PCR amplification conditions, including primer efficiencies, were described previously [19]. Fold changes were obtained using the comparative CT method (ΔΔCt) normalized to *gyrB* expression and compared to WT as a reference.

### 2.14. Statistical Analysis

All data were analyzed using the Statistical Package for Social Sciences (SPSS) version SPSS 26.0 (SPSS Inc., Chicago, IL, USA). Student’s *t*-test and one-way ANOVA were used, and the probability of a difference between groups’ data was greater than 95% to be considered statistically significant.

## 3. Results

### 3.1. Construction of the lmo0159 Deletion Mutant (∆lmo0159)

To clarify the role of the *lmo0159* gene, we inactivated the *lmo0159* gene by introducing an in-frame deletion. The PCR analysis confirmed that, as expected, only a 962 bp band was amplified from the ∆*lmo0159* strain after culturing it in the chloramphenicol-free medium for 25 generations at 30 °C and 30 generations at 37 °C (Figure 2). This indicated that the ∆*lmo0159* strain was genetically stable. 

### 3.2. Impact of Lmo0159 Deletion on L. monocytogenes Growth

To investigate the effect of *lmo0159* on the growth of *L. monocytogenes*, WT and ∆*lmo0159* were incubated at 37 °C for 12 h. *L. monocytogenes* proliferated rapidly from 2–8 h and entered a plateau after 8 h. The growth rate of ∆*lmo0159* was significantly higher than that of WT at 4 h (*p* < 0.0001) (Figure 3). The findings suggest that the *lmo0159* gene is not required for growth of *L. monocytogenes* but may impact growth rate during exponential growth.

### 3.3. Lm0159 Mutant Is Defective in Biofilm Formation

Images of the biofilm surfaces were obtained from an inverted microscope. As shown in Figure 4A, the WT and ∆*lmo0159* strains formed a grid-like biofilm, but the degree of cross-linking and aggregation of the biofilm formed by the WT strain was higher than that of ∆*lmo0159*. The WT strain forms a highly compact biofilm after 48 h, while the biofilm of the deleted strain remains in a lattice structure.

Quantitative biofilm-formation assay results show that the ability of the ∆*lmo0159* strain to form biofilm was not significantly different from that of the WT strain at 8 and 12 h (*p* > 0.05), but the biofilm formation of the ∆*lmo0159* strain was remarkably lower than that of the WT strain at 24 and 48 h (Figure 4B; *p* < 0.01 and *p* < 0.001).

To further determine whether there is a difference between virulence and biofilm, the relative gene expressions of *L. monocytogenes* were analyzed. RT-PCR was performed on *L. monocytogenes* to determine the mRNA expression of fourteen genes, including five biofilm-formation genes, *degU*, *flaA*, *mogR*, *fur,* and *fliP*; and nine virulence genes, *prfA*, *plcA*, *hly*, *mpl*, *actA*, *plcB*, *inlA*, *inlB,* and *inlC*. Compared to the WT strain, the incubation of ∆*lmo0159* resulted in a significant suppression of the transcription of *prfA*, *mpl*, *actA*, *plcB*, *inlB,* and *inlC* (Figure 4C) (*p* < 0.01); and *degU*, *flaA,* and *fur* genes (Figure 4D) (*p* < 0.01), whereas the transcription of *inlA* and *mogR* was upregulated (*p* < 0.01). This suggested that the deletion of *lmo0159* decreased the mRNA expression of virulence and biofilm genes. 

### 3.4. Inhibiting Effects of Lmo0159 on the L. monocytogenes Motility

*L. monocytogenes* spread from the center to the periphery, forming bacterial circles on the plate. The swimming diameter of ∆*lmo0159* was significantly higher than that of WT at 24 h (*p* < 0.05), which was exceptionally significantly higher than that of WT at 36 h and 48 h (*p* < 0.01) (Figure 5A,B). This indicates that the deletion of *lmo0159* promotes the motility of *L. monocytogenes*. *L. monocytogenes* was stab-inoculated into tubes and plates of TSA semi-solid medium and incubated at 25 °C for 48 h. *L. monocytogenes* grew in an inverted umbrella shape around the stab line in the tube, and the growth rate of ∆*lmo0159* was higher than that of the WT at 36 h (Figure 5C). 

To further understand the effect of *lmo0159* on motility at the cellular level, the plaque-formation assay and scratch-healing assay were performed. As shown in Figure 5D, the plaque center appeared punctate agglutination-like, and the plaque was comet-like. There was no significant difference in the number of plaques between the two groups (Figure 5E); the plaque diameter of the WT group was significantly higher than that of the ∆*lmo0159* group in hCMEC/D3 (*p* < 0.001) and L929 (*p* < 0.01) (Figure 5F). It suggests that *lmo0159* promotes the spread of *L. monocytogenes* in hCMEC/D3 and L929. 

### 3.5. Effects of Lmo0159 on Cell Adhesion, Invasion, Intracellular Proliferation, and Virulence Factor Levels

In Figure 6A, the scratch of L929 cells in the three groups showed a healing trend from 0 h to 24 h, and the cells were elongated and fusiform during the migration process. However, the adjacent cells still maintained tight connections. The migration rate of the NC group was significantly higher than that of the WT group and the ∆*lmo0159* group (*p* < 0.0001 and *p* < 0.01, respectively), and that of the WT group was significantly lower than that of the ∆*lmo0159* group (*p* < 0.01; Figure 6B). This suggests that the presence of *lmo0159* reduced the migratory ability of L929 cells.

The ability of WT and ∆*lmo0159* strains to adhere to different eukaryotic cell types was tested. It suggested a decrease in adhesion for the ∆*lmo0159* strain compared to the wild type in these cell lines. In particular, the adhesion rates of ∆*lmo0159* to RAW264.7 and SIEC cells were extremely significantly (*p* ˂ 0.01) lower than those of WT (Figure 6C). The results showed that the cell invasive ability of the ∆*lmo0159* mutant strain was decreased for all cell lines examined (Figure 6D). In addition, the invasive ability of the mutant strain for RAW264.7 and SIEC cell lines was significantly lower than that of the WT strain (*p* ˂ 0.01). To analyze the role of *lmo0159* in other steps of the cell infection process, the intracellular multiplication of the WT and ∆*lmo0159* strains was examined after internalization into mBMECs (Figure 6E), RAW264.7 cells (Figure 6F), SIECs (Figure 6G), and hCMEC/D3 (Figure 6H). The number of viable intracellular bacteria of ∆*lmo0159* infected in mBMECs, hCMEC/D3, and SIECs was significantly lower than that of WT, but RAW264.7 was significantly higher than that of WT. Interestingly, the intracellular multiplication amounts of mutant or wild-type strains in mBMECs, hCMEC/D3, and RAW264.7 were all on the uptrend, but that in SIECs was on the contrary; it suggested that the deletion of *lmo0159* may have a different function in the intestinal epithelium. 

To further explore the mechanism of *lmo0159* in hCMCE/D3 cells challenged with *L. monocytogenes*, the virulence-associated genes were detected by RT-PCR at 12 h after the *L. monocytogenes* challenge. As shown in Figure 6I, nine virulence factors were downregulated to different degrees in the ∆*lmo0159* group, including *inlA* (*p* < 0.05); *hly*, *mpl*, *actA*, *plcB*, and *inlC* (*p* < 0.01); *inlB* (*p* < 0.001); and *prfA* and *plcA* (*p* < 0.0001). The results showed that these nine key virulence factors were jointly involved in the proliferation of *L. monocytogenes* in cells promoted by *lmo0159*.

### 3.6. Inactivation of Lmo0159 Impairs the Virulence of L. monocytogenes

To determine whether Lmo0159 is essential for the virulence of *L. monocytogenes*, in vitro infection studies were performed with the ∆*lmo0159* and WT strains. The plaque formation assay showed that ∆*lmo0159* significantly impaired the ability to spread in hCMEC/D3 compared to the WT strain (Figure 5D). The mouse infection model was then used to evaluate the virulence properties of the ∆*lmo0159* strain. The mean survival time of mice inoculated with ∆*lmo0159* strain (3 days) was significantly longer than that of WT (2 days; *p* < 0.05). The control-group mice had no significant abnormalities or deaths (Figure 7A). The LD_50_ values of WT and Δ*lmo0159* to BALB/c mice were 10^6.22^ and 10^7.19^, respectively (Table 3). The attenuation fold of the Δ*lmo0159* mutant was 9.33. The number of bacteria in the liver, spleen, and brain of BALB/c mice was monitored in mice after intraperitoneal infection with either the ∆*lmo0159* or WT for 72 h. In the spleen, ∆*lmo0159* bacterial counts were significantly lower than those of the WT at 48 and 72 h after infection (*p* < 0.01), and mice infected with the mutant strain carried significantly fewer bacteria in the liver and spleen than the WT strain at 24 h (*p* < 0.05; Figure 7B,C). However, ∆*lmo0159* exhibited viable bacterial counts in the brain that were significantly higher than those of the WT strain (*p* < 0.01; Figure 7D). These results indicate that the *lmo0159* gene contributes to the virulence of *L. monocytogenes*.

To investigate how *lmo0159* drives virulence factors to affect the course of *L. monocytogenes* infection in mice, virulence factors were detected by RT-PCR. In Figure 7E, the virulence factors *hly* (*p* < 0.01), *plcA* (*p* < 0.0001), and *plcB* (*p* < 0.001) of ∆*lmo0159* were significantly decreased in the liver, while *inlC* (*p* < 0.01) was significantly increased. In Figure 7F, the virulence factors of ∆*lmo0159* in brain tissue were significantly increased, including *prfA* (*p* < 0.01); *hly*, *mpl,* and *actA* (*p* < 0.05); and *inlC* (*p* < 0.001). The results showed that *lmo0159* affected the transcription level of virulence factors but showed different trends in different organs of mice. 

## 4. Discussion

*L. monocytogenes* is a Gram-positive facultative anaerobic, non-spore-forming, foodborne pathogenic bacterium widely distributed in nature. It is resistant to adverse environmental conditions and can survive and multiply in human professional and non-professional phagocytes [20,21,22]. The wide variety of cell wall components of Gram-positive bacteria is crucial for bacterial viability [23]. As a type of cell wall protein, the LPXTG surface protein can be anchored to the bacterial cell wall surface by Sort A, which plays a vital virulence role in the infection process and adaptation of bacteria to environmental stress [1,7,24]. 

Comparative genomics studies [25,26] have shown that about one-third of pathogenic Listeria strains carry LPXTG proteins not found in non-pathogenic *Listeria* strains. These results suggest that LPXTG proteins may be related to the pathogenicity of Listeria. However, the biological functions of many LPXTG proteins remain unclear. Lmo0159 is predicted to be an uncharacterized surface-anchored protein with an LPXTG-motif cell wall anchor domain [27,28]. To unravel the contribution of these novel *L. monocytogenes* LPXTG proteins to virulence, the differences in *L. monocytogenes* growth, motility, biofilm formation, cell challenge, and mouse infection were compared between WT and Δ*lmo0159* strains. The results showed that the *lmo0159* gene of *L. monocytogenes* played an essential role in motility, adhesion, invasion, intracellular proliferation, and tissue colonization of various cell models and finally affected the survival ability of infected mice. 

*L. monocytogenes* forms biofilms on many biotic and abiotic surfaces, and this growth mode protects it from several environmental stresses, including desiccation, ultraviolet rays, and chemicals [29,30]. Once formed, the biofilm is difficult to remove, which increases drug resistance and can lead to persistent infection and chronic inflammation in some diseases [31]. In this study, we compared the ability of WT and ∆*lmo0159* strains to form biofilm using morphological observation and quantitative biofilm-formation assay methods. The results showed that the deletion of the *lmo0159* gene significantly reduced the ability to form biofilm. This was consistent with the phenotype. These results suggest that *lmo0159* positively regulates biofilm formation.

Previously published data suggest that flagellar-mediated motility is critical for biofilm formation in *L. monocytogenes* [32,33]. Based on this report, we initially hypothesized that the reduction in biofilm formation caused by the deletion of *lmo0159* might be related to flagellar-mediated motility. Interestingly, the swimming motility of *L. monocytogenes* was increased compared to WT. Therefore, *lmo0159* appeared to positively regulate biofilm formation by interfering with *L. monocytogenes* motility, which is inconsistent with our hypothesis. The gene expression of WT and Δ*lmo0159* strains was analyzed to determine further whether the difference was related to flagellar-mediated motility. The RT-PCR results showed that the deletion of *lmo0159* significantly inhibited the transcription of biofilm-formation genes, flagellar genes, and virulence genes while they were significantly upregulated. Mirzahosseini [34] demonstrates that biofilm-formers are more pathogenic than the planktonic form in urinary tract infection and that biofilm production causes an increase in the pathogenicity of *uropathogenic E. coli* isolates. Our results showed that the deletion of *lmo0159* downregulated most of the virulence genes, as well as the genes involved in biofilm formation (*degU*, *flaA,* and *fur*) and upregulated the flagellar gene transcriptional repressor *mogR* under biofilm state. This is consistent with the biofilm phenotype and the results of Mirzahosseini.

The deletion of *lmo0159* significantly reduced the efficiency of *L. monocytogenes* adhesion and invasion of RAW264.7 and SIECs and showed a downward trend in intracellular proliferation of mBMECs, hCMEC/D3, and SIECs. These results were consistent with the reduction in bacterial colonization in liver and spleen tissues of Δ*lmo0159* mice, which ultimately promoted the survival of Δ*lmo0159* mice and was attenuated by 9.33-fold compared to WT mice. *L. monocytogenes* virulence factors are involved in the different steps of the cell infection cycle [4]. The LPXTG surface proteins, such as InlA, LapB, InlF, and P60, are involved in adhesion and invasion in several cell lines [6]. Among them, InlA is the first Listeria LPXTG protein to be identified and plays a crucial role in crossing the intestinal and placental barriers [35,36,37]. Lmo0171, an LPXTG surface protein, and the disruption of the *lmo0171* gene resulted in the decreased ability to invade Int407, Hep-2, and HeLa cell lines and decreased adhesion efficiency to Int407 cells [38]. Our results are consistent with the finding of Reference [38], suggesting that the *lmo0159,* which also codes LPXTG protein, contributes to the virulence of *L. monocytogenes*. Furthermore, the decrease in virulence in the Δ*lmo0159* group was not due to the reduction of *L. monocytogenes* growth and motility but to the interaction of *lmo0159* with cells and mice. This further confirmed that *lmo0159* was closely related to virulence.

The deletion of *lmo0159* attenuated the motility of *L. monocytogenes* in hCMEC/D3 cells and reduced scratch damage in L929 cells. In addition to virulence, *prfA* and other genes are also involved in biofilm formation. Studies have shown that InlA and InlB, two major proteins in *L. monocytogenes* biofilm, play an essential role in initial bacterial adhesion [39]. Loss of *prfA* dramatically alters gene expression patterns in *L. monocytogenes* biofilms and reduces biofilm-formation capacity [40]. Our mRNA levels are consistent with this. Except for *inlA*, genes regulated by PrfA have protein expression capacity that varies with the concentration of the PrfA regulator and its affinity to the promoter. Although *inlA* is regulated by PrfA, the expression level of PrfA-dependent genes may differ due to the different affinity of the promoter for RNA polymerase and the structure of the 5′ untranslated region [41]. In the early stage of biofilm formation, bacteria adhere to the surface of solid media by motility, thereby forming a biofilm structure. Bacterial motility was positively correlated with biofilm formation [42]. The increased growth and motility of *L. monocytogenes* in the Δ*lmo0159* group did not affect the significant decrease in biofilm-formation ability, which further proved the positive regulatory effect of *lmo0159* on biofilm formation and virulence.

Our mouse infection results show that the absence of *lmo0159* can significantly reduce WT virulence in mice and reduce the ability of the bacteria to colonize the liver and spleen of mice. These results indicate that *lmo0159* is essential for virulence in mice. Interestingly, higher numbers of mutant bacteria were found in the brain than the WT, suggesting that *lmo0159* may play a role in crossing the blood–brain barrier.

This result contradicts the intracellular proliferation results of Δ*lmo0159* in hCMEC/D3 and mBMEC. This may be because the cell type in a two-dimensional culture system is relatively single, and the interaction between cells is very different from the natural tissue. The single-cell models of hCMEC/D3 and mBMEC cannot fully simulate the complex functional characteristics of the brain in vivo. Brain organoids can provide a new research model to study the development of brain diseases. Although brain organoids can simulate the cellular, molecular, and functional characteristics of brain development, they cannot be implemented in our laboratory due to the technical difficulty of long-term healthy culture. The pathways by which *L. monocytogenes* crosses the blood–brain barrier can be divided into two categories: direct infection of brain microvascular endothelial cells and invasion of monocytes-macrophages (“Trojan horse” pathway) [43]. The Trojan horse form means that after *L. monocytogenes* is engulfed by phagocytes (monocytes, dendritic cells, and neutrophils), the infected phagocytes transfer bacteria to the brain via the bloodstream, causing infection. In this study, *lmo0159* may play an essential role in the passage of *L. monocytogenes* through the blood–brain barrier in the form of a Trojan horse, resulting in the high number of bacteria in the brain of the Δ*lmo0159* group. The increased proliferation trend of Δ*lmo0159* in the mononuclear-macrophage RAW264.7 in Figure 6F supports this possibility, and the pathways and mechanisms of *lmo0159* entry into the brain warrant further investigation.

For logistical reasons, genetic complementation of the deletion mutant was not pursued in the current study. Restoration of the observed phenotypes upon genetic complementation would be important for accurate assessments of the roles of *lmo0159* in virulence, motility, and biofilm formation. 

## 5. Conclusions

In summary, our study demonstrated that a *lmo0159* gene deletion mutant was significantly attenuated in virulence. Deletion of *lmo0159* promoted swimming motility and decreased biofilm-formation ability in vitro. ∆*lmo0159* suppressed the *L. monocytogenes* adhesion, invasion, proliferation, and motility, while promoting healing of the cell scratch; increased the LD_50_ and survival time in mice; and reduced the *L. monocytogenes* reproduction in liver and spleen compared with WT. Further studies showed that the deletion of *lmo0159* resulted in the downregulation of gene expression related to virulence and biofilm formation. The construction of a genetically complemented strain and the restoration of the deletion phenotypes would provide further evidence on the roles of this novel determinant. Thus, for the first time, we have identified *lmo0159* as a novel virulence factor among *L. monocytogenes* LPXTG proteins, which may contribute to biofilm formation and virulence. Taken together, these findings provide fertile ground for future investigations.

## Figures and Tables

**Figure 1 microorganisms-12-01354-f001:**
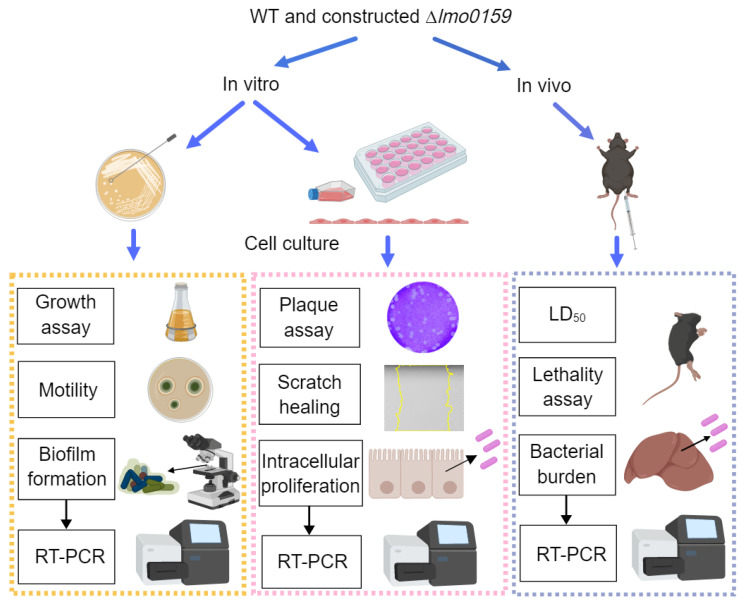
Experimental design. Experiments were performed both in vivo and in vitro, as indicated. In vitro assessments included growth, motility, biofilm-formation measurements, and RT-PCR expression of related genes in the biofilm state. The efficiency of adhesion, invasion, and intracellular proliferation was also evaluated. In vivo experiments were performed in LD_50_, bacterial load, lethality assay, and expression of virulence genes, including assessments of mouse survival rate and virulence levels of *L. monocytogenes*.

**Figure 2 microorganisms-12-01354-f002:**
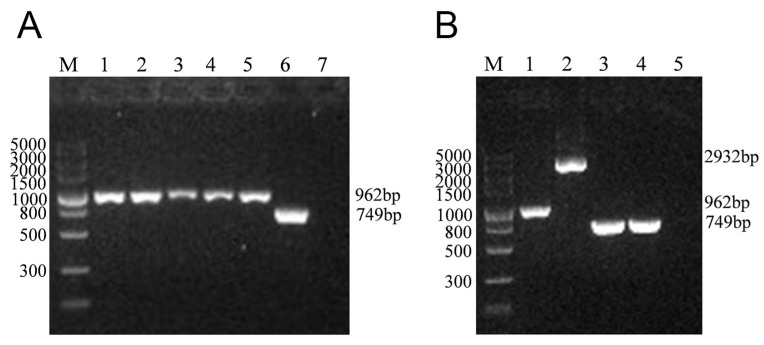
Construction of the deletion mutant strain ∆*lmo0159*. (**A**) Detection of genetic stability of ∆*lmo0159* using primers 0159-A and 0159-B (962 bp) and simultaneous identification of *L. monocytogenes* using hly-F and hly-R (749 bp) primers. (**B**) Identification of WT (2932 bp) and ∆*lmo0159* (962 bp) using primers 0159-A and 0159-B and simultaneous identification of *L. monocytogenes* using primers from *hly*-F and *hly*-R (749 bp).

**Figure 3 microorganisms-12-01354-f003:**
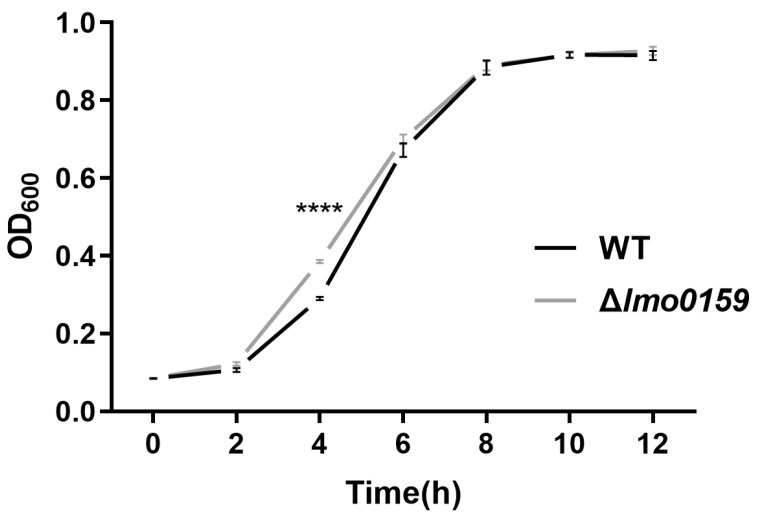
*L. monocytogenes* was incubated in BHI medium, and the growth rates were measured at 37 °C. Values represent the mean ± SD (n = 3). **** *p* < 0.0001. If the error bar is shorter than the symbol’s size, the software simply will not draw it, even if the symbol is clear. Thus, the size of the symbols in the legend is greatly reduced to show rather small error bars clearly.

**Figure 4 microorganisms-12-01354-f004:**
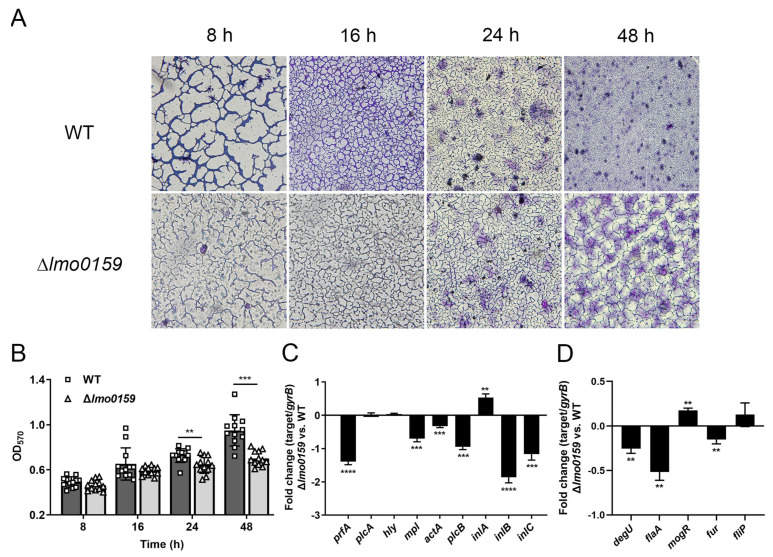
Deletion of *lmo0159* reduced the ability of *L. monocytogenes* to form biofilm. (**A**) Micrographs of biofilm formed by WT and ∆*lmo0159* strain photographed with an inverted microscope (400×). (**B**) Biofilm development by WT and ∆*lmo0159* strains in 96-well microtiter plates in a crystal violet staining assay. The average OD_570_ of crystal violet measured quantitative biofilm formation. The fold change in virulence genes (**C**) and biofilm genes (**D**) at 48 h was determined by RT-PCR. Values represent the mean ± SD (n = 3). ** *p* < 0.01, *** *p* < 0.001, and **** *p* < 0.0001.

**Figure 5 microorganisms-12-01354-f005:**
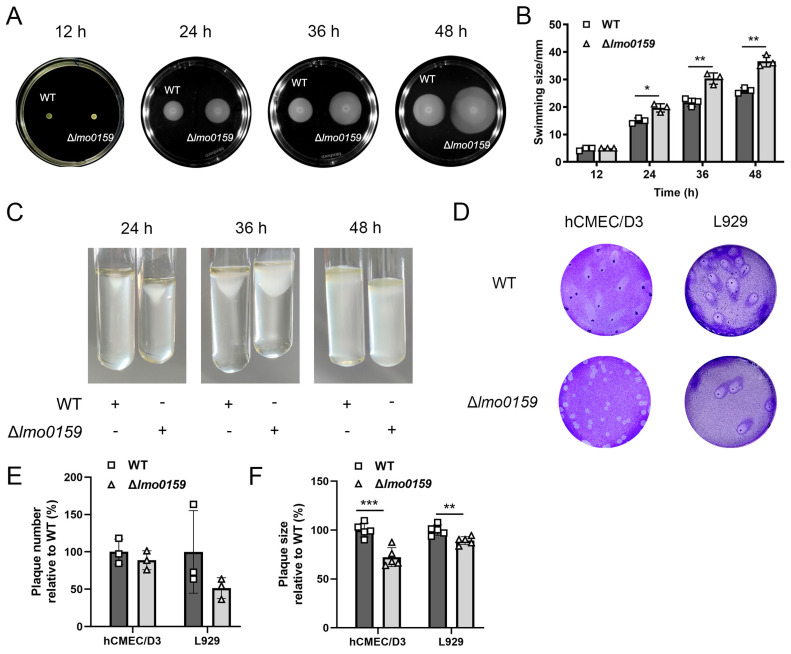
Determination of the effect of *lmo0159* on the motility of *L. monocytogenes*. (**A**) Plate motility assay. *L. monocytogenes* stab inoculated TSA semi-solid medium plates were observed. (**B**) Quantification of motility in TSA plates. Values represent the mean ± SD (n = 3). (**C**) Umbrella motility assay. TSA semi-solid medium tubes stabbed with *L. monocytogenes* were observed. Plus sign (+) represents the inoculation of *L. monocytogenes* in the same row in the tube, while minus sign (−) does not. (**D**) Plaque assay. Movement circles formed by *L. monocytogenes* on hCMEC/D3 and L929 after 3 d were visualized by crystal violet staining (2×). (**E**) Quantification of plaques number. (**F**) Measurement of plaque diameter. * *p* < 0.05, ** *p* < 0.01, and *** *p* < 0.001.

**Figure 6 microorganisms-12-01354-f006:**
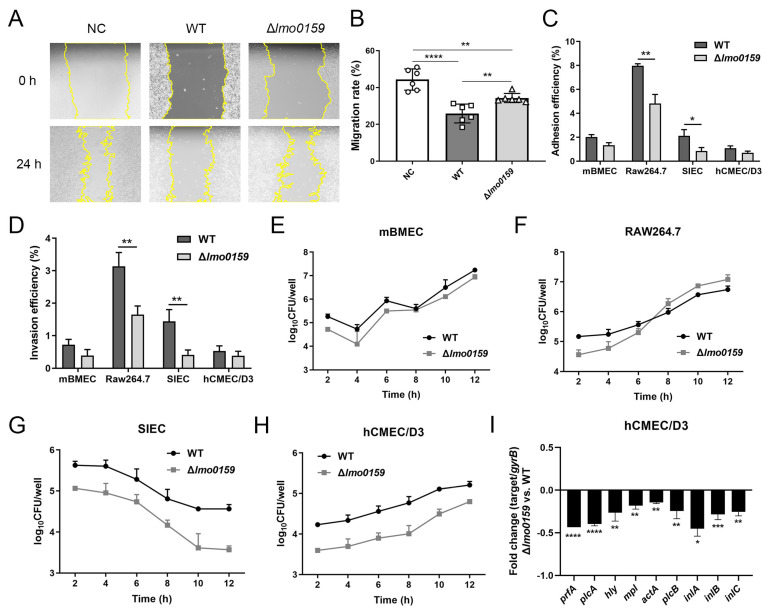
Role of *lmo0159* in the challenge of cells by *L. monocytogenes*. (**A**) Scratch-healing assay. Changes in the motility of L929 after challenge with *L. monocytogenes* were examined by scratching (100×). (**B**) Quantification of migration rate of L929. Values represent the mean ± SD (n = 6). (**C**) The adhesion rate of WT and ∆*lmo0159*. (**D**) The invasion rate of WT and ∆*lmo0159*, respectively. The intracellular proliferation number of viable bacteria in mBMEC (**E**), RAW264.7 (**F**), SIEC (**G**), and hCMEC/D3 (**H**). The fold change in *L. monocytogenes* virulence genes in hCMEC/D3 at 12 h post-challenge was determined by RT-PCR (**I**). * *p* < 0.05, ** *p* < 0.01, *** *p* < 0.001, and **** *p* < 0.0001.

**Figure 7 microorganisms-12-01354-f007:**
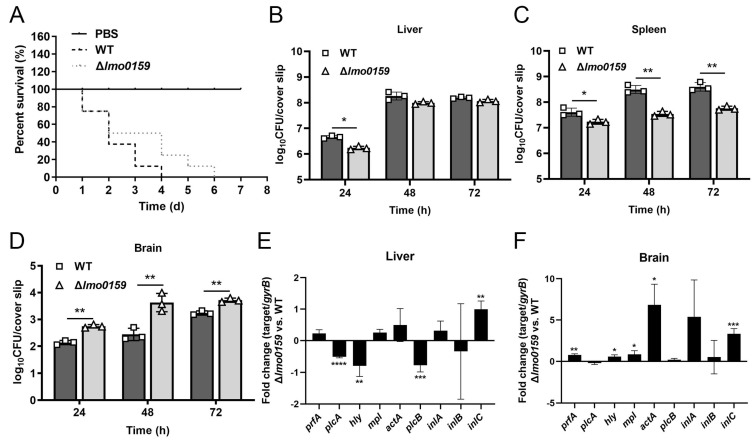
The effect of *lmo0159* on the virulence level was evaluated at the mouse level. (**A**) Mouse lethality assay. Eight-day survival curves of sixteen mice were calculated (*p* < 0.05; Kaplan–Meier test). Bacterial counts were monitored in the liver (**B**), spleen (**C**), and brain (**D**) of mice at 24 h, 48 h, and 72 h post-challenge (significance by Mann-Whitney test). The fold change in *L. monocytogenes* virulence genes in liver (**E**) and brain (**F**) at 48 h post-challenge was determined by RT-PCR. Values represent mean ± SD (n = 3). * *p* < 0.05, ** *p* < 0.01, *** *p* < 0.001, and **** *p* < 0.0001.

**Table 1 microorganisms-12-01354-t001:** List of primers used for PCR.

Primer	Nucleotide Sequence (5′-3′)	Target Gene	Length (bp)
0159-F1	CGGGATCCCAAATGCTCATTCTGGACC (*BamH* I)	Upstream sequence of *lmo0159* gene (A1)	388
0159-R1	TCGTTTCAGCAGAAGACCAGATTGTGCTAGCGACGATT		
0159-F2	AATCGTCGCTAGCACAATCTGGTCTTCTGCTGAAACGA	Downstream sequence of *lmo0159* gene (A2)	409
0159-R2	AACTGCAGTCGTAATAACCGAAGTCCG (*Pst* I)		
0159-A	TGAGTATGCTGGTATGGGA	Verify primers of *lmo0159* deletion	2932/962
0159-B	GTCGCCTTCCTTCACATTA		
*hly-F*	CTGAATTCGGCTGTTACTAAAGAGCAGTTGC	*hly*	749
*hly-R*	ATGGATCCTTAGCCCCAGATGGAGATATTCTA		

**Table 2 microorganisms-12-01354-t002:** Target gene primer sequences used for RT-PCR.

Gene	forward Primer (5′-3′)	Reverse Primer (5′-3′)
*degU*	ACGCATAGAGAGTGCGAGGTATT	CCCAATTCCGCGGTTACTT
*flaA*	CTGGTATGAGTCGCCTTAG	CATTTGCGGTGTTTGGTTTG
*mogR*	AACTGCCGAAGAAATCTACCATTT	CGATTCCACCGTGTTCTTCA
*fur*	ACAGTTGCAAGGCCAGTGTCGGGTG	TGGAAGGTCGTATTGGACGCATTAA
*fliP*	TTGGCCGGGTGTGAATGT	CCATTTACACCAAGCGAATCC

**Table 3 microorganisms-12-01354-t003:** Calculation of LD_50_ of mice challenged with WT and Δ*lmo0159*.

Strain	Group	Does	No. of Animals	Death No.	Mortality	LD_50_ Median Lethal Dose
WT	1	5.3 × 10^9^	8	8	1	10^6.22^
	2	5.3 × 10^8^	8	8	1	
	3	5.3 × 10^7^	8	7	0.875	
	4	5.3 × 10^6^	8	6	0.75	
	5	5.3 × 10^5^	8	3	0.375	
Δ*lmo0159*	1	3.7 × 10^9^	8	8	1	10^7.19^
	2	3.7 × 10^8^	8	7	0.875	
	3	3.7 × 10^7^	8	5	0.625	
	4	3.7 × 10^6^	8	3	0.375	
	5	3.7 × 10^5^	8	0	0	

## Data Availability

The original contributions presented in the study are included in the article, further inquiries can be directed to the corresponding authors.

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
