# Peer review of "Biofilm Formation, Motility, and Virulence of *Listeria monocytogenes* Are Reduced by Deletion of the Gene *lmo0159*, a Novel Listerial LPXTG Surface Protein"

_microorganisms, 2024, doi:10.3390/microorganisms12071354_

Round 1

Reviewer 1 Report

Comments and Suggestions for Authors

The authors studied the impact of a new Listerial LPXTG surface protein, called lmo0159, on the Lm virulence and biofilm creation. They demonstrated a notable lmo0159 gene deletion mutant decline in virulence with a substantial boost in the migration rate and bacterial growth in the brain. In addition, they reported that the lmo0159’s deletion reduced cellular responses of genes related to virulence and biofilm development.

There are a few minor notices that need to be addressed:

·         The study is valuable, but the abstract is very weak and doesn’t show the actual value of the research. So, I would highly recommend rewriting the abstract.

·         There is a lot of information in the figures’ captions; some of this information should be addressed in the methods sections, and some should be moved to the results sections.  

·         Some grammar and punctuation issues were noticed. 

Comments on the Quality of English Language

Some grammar and punctuation issues were noticed. 

Author Response

Dear reviewer,

Thank you very much for your comments and professional advice. These opinions help to improve academic rigor of our article. Based on your suggestion and request, we have made corrected modifications on the revised manuscript. The detailed corrections are listed below:

  1. question: The study is valuable, but the abstract is very weak and doesn’t show the actual value of the research. So, I would highly recommend rewriting the abstract.

Response: The previous abstract was quite weak and has been rewritten.

  1. question: There is a lot of information in the figures’captions; some of this information should be addressed in the methods sections, and some should be moved to the results sections.

Response: Excessive details in the image description have been removed or transferred to methods such as Fig7: ’’The mice were inoculated intraperitoneally, the number of deaths was recorded, and the corresponding eight-day survival curves were calculated.’’ has been moved to Method 2.12.

  1. question: Some grammar and punctuation issues were noticed.

Response: This has been checked and revised in the full text.

We appreciate the reviewer's warm work earnestly and hope the corrections will meet with approval. Once again, thank you for your comments and suggestions.

Sincerely,

Weidi Shi

Reviewer 2 Report

Comments and Suggestions for Authors

The research submitted by Shi et al. is very interesting, as the authors were able to detect a new virulence factor of the bacterium Listeria monocytogenes. This bacterium is an important foodborne pathogen and can cause severe, highly lethal diseases in immunosuppressed individuals. Therefore, understanding its virulence mechanisms is of utmost importance and highlights the significance of the authors' findings.

The introduction is well-written and provides the necessary understanding of the study. The methodology seems adequate; however, NO bibliographic references were included, which prevents a proper evaluation of the techniques employed. This requires a complete revision of the section, which should be thoroughly referenced to allow for reproducibility and repeatability. This will also enable a better assessment of the conducted research. Additionally, this section needs a subtopic clearly describing the experimental design used. The results, discussion, and conclusion sections are adequate and require few adjustments.

Remarks:

The authors should standardize the correct taxonomic usage of the bacterium throughout the manuscript. The first mention in the text should be as “Listeria monocytogenes” and subsequently as “L. monocytogenes”. “Lm” is entirely inappropriate for use in a scientific article. Standardize this throughout the manuscript.

In the Keywords section, do not repeat words already present in the title. Use other keywords that contextualize the study area for better indexing.

In the Materials & Methods section, the authors should include an initial subtopic describing the general experimental design used. I suggest including a flowchart to allow for better understanding. In the current format, starting with strains and primers, it is impossible to follow a logical line of understanding of the article. It must first be clear what will be conducted through the presentation of an appropriate design before mentioning the strains and primers used in PCR.

In the Materials & Methods section, there are practically no bibliographic references for the techniques employed. The authors should ensure to adequately include bibliographic references for each technique used in the research.

The titles of all tables and figures should be revised to be self-explanatory and understandable, so they can be comprehensible even separately from the article. Provide detailed information on what each figure or table aims to present.

L66 – I don't understand. What do the authors mean by "produced"? What are they trying to state?

L104 – Be clear about "small amount". This is a scientific article, and the techniques employed should be reproducible.

L124 – How was the "mental state" evaluated? Be precise and clear.

The primers in section 2.10 should be presented in a table. Additionally, standardize so that all genes appear in italics throughout the manuscript (not just in this section).

L204 – State that the significance level of the study was 95%, not that p<0.05 was significant.

L402-412 – Include references.

Author Response

Dear reviewer,

Thank you very much for your comments and professional advice. These opinions help to improve academic rigor of our article. Based on your suggestion and request, we have made corrected modifications on the revised manuscript. The detailed corrections are listed below:

1.question: The authors should standardize the correct taxonomic usage of the bacterium throughout the manuscript. The first mention in the text should be as “Listeria monocytogenes” and subsequently as “L. monocytogenes”. “Lm” is entirely inappropriate for use in a scientific article. Standardize this throughout the manuscript.

Response: The Lm in the full text has been replaced with L. monocytogenes.

2.question: In the Keywords section, do not repeat words already present in the title. Use other keywords that contextualize the study area for better indexing.

Response:  The previous keyword ‘’Lm; lmo0159; deletion; Biofilm formation; Virulence’’ has been changed to ‘’foodborne pathogen; L. monocytogenes; deletion mutant; bacteria-host interactions; RT-PCR’’.

3.question: In the Materials & Methods section, the authors should include an initial subtopic describing the general experimental design used. I suggest including a flowchart to allow for better understanding. In the current format, starting with strains and primers, it is impossible to follow a logical line of understanding of the article. It must first be clear what will be conducted through the presentation of an appropriate design before mentioning the strains and primers used in PCR.

Response: The experimental design was added in Method 2.3. and a schematic diagram of the process was made in Fig 1. Experimental design.

4.question: In the Materials & Methods section, there are practically no bibliographic references for the techniques employed. The authors should ensure to adequately include bibliographic references for each technique used in the research.

Response: The original test methods and techniques literature was searched and cited. These cite located in L159, L167, L175, L185, L197, L215, L235, L243, and L252.

5.question: The titles of all tables and figures should be revised to be self-explanatory and understandable so they can be comprehensible even separately from the article. Provide detailed information on what each figure or table aims to present.

Response: The titles of tables and figures have been modified.

6.question: L66 – I don't understand. What do the authors mean by "produced"? What are they trying to state?

Response: "produced" is a wrong word to use; it has been amended to cryopreserved (L82).

7.question: L104 – Be clear about "small amount". This is a scientific article, and the techniques employed should be reproducible.

Response: The front quarter of a sterilized toothpick was dipped into the bacterial solution to ensure that the initial concentration was consistent (L170).

8.question: L124 – How was the "mental state" evaluated? Be precise and clear.

Response: Because we did not score the mental state of the mice, ‘’mental state’’ was removed (L221).

9.question: The primers in section 2.10 should be presented in a table. Additionally, standardize so that all genes appear in italics throughout the manuscript (not just in this section).

Response: Primer is in the form of a table to show (L254), the full-text gene is italic.

10.question: L204 – State that the significance level of the study was 95%, not that p<0.05 was significant.

Response: It has been stated (L265).

11.question: L402-412 – Include references.

Response: References were added for L441 and L446.

We appreciate the reviewer's warm work earnestly and hope the corrections will meet with approval. Once again, thank you for your comments and suggestions.

Sincerely,

Weidi Shi

Manuscript is in the attachment and modification cover letter is at the end of the article.
